# Comparative Analysis of Skeletal Changes, Occlusal Changes, and Palatal Morphology in Children with Mild Class III Malocclusion Treated with Elastodontic Appliances and Bimaxillary Plates

**DOI:** 10.3390/children10071219

**Published:** 2023-07-14

**Authors:** Vincenzo Ronsivalle, Vincenzo Quinzi, Salvatore La Rosa, Rosalia Leonardi, Antonino Lo Giudice

**Affiliations:** 1Department of Medical-Surgical Specialties—Section of Orthodontics, School of Dentistry, University of Catania, Policlinico Universitario “G. Rodolico-San Marco”, Via Santa Sofia 78, 95123 Catania, Italy; vincenzo.ronsivalle@hotmail.it (V.R.); salvo.larosa11@lve.it (S.L.R.); rleonard@unict.it (R.L.); 2Department of Life, Health & Environmental Sciences, Postgraduate School of Orthodontics, University of L’Aquila, 67100 L’Aquila, Italy; vincenzo.quinzi@uniqva.it

**Keywords:** class III malocclusion, children, interceptive orthodontics, digital orthodontics, early treatment

## Abstract

Background: The aim of the present study was to compare the changes observed in children after the early treatment of mild class III malocclusion using bimaxillary removable plates supported by class III elastics and elastodontic devices. Methods: Twenty children (mean age 7.6 ± 1.1 years) with signs of class III malocclusion were treated using by-maxillary plates (PG group) with class III elastics (10 subjects = mean age 7.9 ± 1.3 years) or using class III elastodontic devices (EG group) (10 subjects = mean age 7.4 ± 0.8 years). Digital models and lateral cephalograms were obtained before treatment (T0) and at the end of treatment (T1). The digital models were analyzed to assess occlusal changes and maxillary morphology using the surface-to-surface matching technique. Changes in cephalometric parameters were also analyzed. The data outcomes were statistically analyzed using the paired Student’s *t* test for inter-timing assessments and the independent Student’s *t* test for inter-group assessments. Results: Both groups showed correction of class III malocclusions, with a significant increase in the ANB angle and the overjet (*p* < 0.05). Subjects in the PG group exhibited a greater reduction in the inter-incisal angle compared to the EG group (*p* < 0.05). The children in the EG group had a significantly lower percentage of palatal morphology matching between T0 and T1 compared to the PG group (*p* < 0.05), suggesting greater morphological changes in the palate. Conclusions: Elastodontic appliances (EAs) and bi-maxillary plates successfully correct class III malocclusions in children. However, elastodontic devices significantly improved the morphology of the palate, both in the transverse and anteroposterior directions.

## 1. Introduction

Early intervention for anterior crossbite is advocated due to its detrimental effects on maxillofacial growth and function, including the potential development of temporomandibular joint disorder [1,2,3,4]. The primary objective of interceptive treatment for class III malocclusion is to alleviate the anterior crossbite by restoring incisal guidance and, as a secondary outcome, to minimize the risk of relapse [5], while maintaining mandibular constraint by the upper arch during growth. Various appliances such as chin cups, face masks, functional devices, intra-oral plates, and, more recently, elastodontic devices can be employed for early treatment in deciduous or mixed dentition. Each method has its advantages and disadvantages, and the appropriate treatment modality should be selected based on accurate diagnosis and the severity of the malocclusion. However, early treatment of anterior crossbite does not exclude the possibility of later interventions to control skeletal growth [6]. In mild cases of early class III malocclusion, intra-oral devices such as maxillary plates, bi-maxillary plates (with or without class III elastics), and functional appliances are preferred over extra-oral orthopedic appliances, which are more suitable for severe skeletal class III cases or as a second-stage treatment in mixed dentition.

Elastodontic treatment represents an interceptive approach utilizing removable elastomeric appliances that exert light and biologically compatible forces. These forces facilitate the correction of malocclusions during the developmental stage, through a combined action of providing guidance for active and passive tooth eruption and vestibular flanges that minimize the influence of perioral muscles on tooth position and arch development [7]. Elastodontic appliances are available in various designs tailored to address different malocclusions, including class III cases. Despite the increasing usage of elastodontic appliances among orthodontists and pediatric dentists, there is a scarcity of evidence regarding their effectiveness, emphasizing the need for further clinical studies on this subject [8]. Specifically, there is currently no evidence regarding the efficacy and mechanisms of elastodontic appliances in correcting mild class III malocclusion with anterior crossbite in children. The objective of this study was to assess occlusal changes and maxillary morphology following the treatment of anterior crossbite using bi-maxillary plates and an elastodontic appliance designed for class III correction in a retrospective cohort of children in the early mixed dentition stage. To achieve this, a specific 3D imaging technology involving the superimposition of pre-treatment and post-treatment maxillary intra-oral scans was utilized to evaluate morphological alterations in the palate throughout the treatment phases. The null hypothesis was the absence of significant differences between changes in the palatal morphology and occlusal parameters that occurred after treatment with elastodontic appliances and bimaxillary plates.

## 2. Materials and Methods

### 2.1. Study Sample

This retrospective study was carried out following the Helsinki Declaration on medical protocols and ethics and was approved by the local institutional review board (protocol n. 119/2019/po-Q.A.M.D.I). An informed consent form for orthodontic treatment and for research purposes was signed by the parents of all of the included subjects. The sample of this retrospective study included 20 children (mean age 7.6 ± 1.1 years) with class III malocclusion seeking orthodontic treatment at an orthodontic private practice in Catania between September 2016 and March 2023.

The inclusion criteria were as follows: (1) functional anterior crossbite, defined as a test-to-test interincisal relationship in centric occlusion with an anterior mandibular shift and (2) lateral cephalograms and digital models recorded before treatment (T0) and after one year of therapy (T1). The exclusion criteria were missing teeth, temporomandibular sound or pain, previous orthodontic treatment, carious lesions, mobility of deciduous posterior teeth, and craniofacial deformities. Ten subjects (mean age 7.9 ± 1.3 years) underwent treatment using bi-maxillary plates with class III elastics, while ten subjects (mean age 7.4 ± 0.8 years) were treated using a class III elastodontic mono-block appliance AMCOP Class III activator (Ortho Protec, Bari, BA, Italy).

A preliminary calculation of the sample size power was performed on subjects satisfying the inclusion/exclusion criteria. The analysis showed that 10 subjects for each group were required to detect a mean difference of 14.3% of T0-T1 surface agreement between the EG and PG groups, with a power of 80% and a significance level of 0.05.

### 2.2. Treatment

In both groups, the clinical outcome established before treatment was the correction of the anterior crossbite and the establishment of a physiological sagittal and vertical incisor relationship, which would mitigate the tendency to the growth pattern of skeletal class III [5]. In this regard, the bi-maxillary plates (PG group, Figure 1A) with class III elastics were used with the aim of protracting the maxillary arch while controlling the sagittal projection of the lower arch. Instead, the class III EAs (EG Group, Figure 1B) feature lingual flanges producing forces that promote ventral pressure on the premaxilla and reaction forces that counteract mandibular growth. Furthermore, the presence of vestibular shields isolates the dento-skeletal structures from occlusal and/or muscular components that could interfere with basal growth. In both groups, the subjects were instructed to wear the appliance at night and for two hours during the day. In the EG group, the children were instructed to bite the device during daily wear, keeping the lips in contact. Once the overjet was corrected, children were recommended to maintain the appliance for only two hours per day. Alginate impressions were taken before treatment (T0) and after 12 months (T1) and digitalization of plaster models was performed using a D2000 3D desktop scanner (3Shape, Copenhagen, Denmark) for the purpose of the present study.

### 2.3. Measurements

All T0 and T1 digital models were imported into 3-Matic software (vr. 13, Materialise NV, Leuven, Belgium) and linear measurements of overbite and overjet were performed (Figure 2A,B).

A specific 3D imaging technology was used to perform the surface analysis between the pre-treatment and post-treatment palatal morphology. The procedures involve the following steps:

(a) Model superimposition (3-matic Medical software (vr. 13, Materialise NV, Leuven, Belgium)). We determined the definition of the median palatal plane for each maxillary model and the superposition of the digital anatomies of the palate between T0 and T1 using this plane as a reference (Figure 3A–C). The median palatal plane (MPP) was delineated by connecting two anatomical landmarks identified along the median palatal raphe and highlighted in the color red. The initial landmark indicated the location on the median palatal raphe adjacent to the second ruga. The second landmark was positioned on the median palatal raphe, precisely 1 cm distal to the first landmark.

(b) Segmentation of the palate (Meshmixer 3.1.373 software; Autodesk, San Rafael, CA, USA). A three-dimensional (3D) model that excluded the alveolar process (including dentition) was created by generating a gingival plane that passed through the most apical points of the dento-gingival junction of all teeth, from the right first molar to the left first molar (Figure 3D–E).

(c) Deviation analysis (Geomagic Control X software; 3D Systems, version 2018.1.1, 3D Systems, Rock Hill, SC, USA). The evaluation of palatal changes between T0 and T1 was performed using deviation analysis. This analysis automatically calculated linear distances between homolog points of the two palatal models measured across 100% of the surface points. The distances between the surface points of the two superimposed models were converted to root mean square (RMS). Additionally, 3D color-coded maps were generated with a tolerance range of 0.5 mm (shown in green) to visualize and identify any discrepancies between the model surfaces (Figure 4). The percentages of distance values falling within the tolerance range were calculated. These percentages provided quantitative data on the extent of correspondence between the original and mirrored models, thereby offering information about the morphological characteristics of the palate at both T0 and T1.

Cephalometric analysis was performed using Dolphin Imaging Software, (Dolphin Imaging, version 11.0, Chatsworth, CA, USA) on the T0 and T1 lateral cephalograms and the following skeletal and dentoalveolar parameters were evaluated: SNA^, SNB^, ANB^ and inter-incisal angle. In those children featuring anterior occlusal interferences with forward shift of the mandible, a lateral cephalogram was performed in centric relation using a wax jig (Figure 5A,B).

All the procedures, including model superimposition, segmentation, deviation analysis, and cephalometric analysis were performed by an experienced operator with 10 years of expertise in clinical and digital orthodontics (V.R.). To assess intra-operator reliability, the same operator repeated the measurements four weeks later. Furthermore, a second expert operator (A.L.G.) performed the digital workflow to evaluate inter-operator reliability.

### 2.4. Statistical analysis

Descriptive statistics were utilized to examine the demographic and clinical characteristics of the PG and EG. The Student’s *t*-test and chi-square test were employed to compare numerical (age) and categorical (gender and skeletal maturity) characteristics between the TG and CG.

The preliminary data analysis involved conducting the Shapiro–Wilk test to assess the data distribution and Levene’s test to evaluate the equality of variance. Since the data exhibited a normal distribution, parametric tests were employed. The paired Student’s *t*-test was used for inter-timing comparison of the tested parameters, while the independent Student’s *t*-test was used for inter-group comparisons.

Intra-examiner reliability was evaluated using the intraclass correlation coefficient (ICC). The statistical analysis was performed using SPSS^®^ version 24 Statistics software (IBM Corporation, 1 New Orchard Road, Armonk, NY, USA).

## 3. Results

Table 1 reports the demographic characteristics assessed via statistical analysis. No differences were recorded between the two groups in terms of the subjects’ distribution according to age, sex, and skeletal maturation.

Both groups showed complete correction of sagittal skeletal relationships, as indicated by an increase in the SNA and ANB angles and a decrease in the SNB angle (*p* < 0.05), with no significant difference between the two groups (*p* > 0.05) (Table 2). The increase in the SNA angle did not reach statistical significance in both group (*p* > 0.05). However, subjects in the PG group exhibited a greater reduction in the inter-incisal angle compared to the EG group (*p* < 0.05) (Table 2).

Furthermore, both groups demonstrated a significant improvement in their overjet (*p* < 0.05), with values approaching the normal range (Table 2). Data on overbite are not reported in the tables, since many feature incomplete eruption of central/lateral incisors. Lastly, subjects in the EG group had a lower percentage of palatal morphology matching between T0 and T1 compared to the PG group (EG = 70.12% ± 2.95; PG = 84.51% ± 3.03) (*p* < 0.05) (Table 3). In both groups, the area of mismatch between T0 and T1 encompassed the entire region of the palate, both in the cranio-caudal and transverse directions (Figure 4). In the EG group, an area of mismatch was observed in the anterior retro-incisive region, while in both groups, the region corresponding to the second and third palatal rugae showed no significant changes.

## 4. Discussion

Interceptive treatment of class III malocclusion offers several advantages. It can help prevent the progression of the malocclusion, minimize the need for more invasive treatments, and improve the patient’s appearance, self-esteem, and overall oral health [9,10]. However, the success of interceptive treatment relies on early diagnosis and appropriate treatment timing. The correction of class III in children is generally performed using orthopaedic/functional appliances. Such devices represent the gold standard for the skeletal control of the altered sagittal growth pattern between the maxilla and the mandible [11]. However, children presenting mild class III or pseudo-class III could benefit from simple and less invasive intra-oral devices that can correct the sagittal inter-maxillary discrepancy with less discomfort compared to extra-oral appliances [5]. In this regard, intra-oral appliance could represent a first line of intervention for treating children with mild class III, without excluding further intervention with orthopaedic appliances (children), orthodontic appliances (adolescents), and or orthognathic surgery (adults).

The use of elastodontic appliances (EAs) for interceptive orthodontic treatment in growing patients is becoming more widespread among orthodontists and pediatric dentists [12,13,14,15]. However, there is currently limited evidence available to support the use of elastodontic therapy in mixed dentition, with only a few case reports and retrospective studies. To the best of our knowledge, this study is the first of its kind to assess the efficacy of elastodontic appliances in the early correction of mild class III malocclusion in children when comparing it to the conventional approach that involves the use of bi-maxillary plates supported by class III elastics.

Both treatment approaches tested in the present study determined the achievement of the established clinical outcome, i.e., the correction of the negative overjet and the re-establishment of physiological mandibular function. In this regard, considerable evidence-based literature suggests that early treatment using simple appliances can restore abnormal growth patterns, leading to a stable occlusion after treatment [5,16].

Concerning skeletal changes, in both groups, there was a significant improvement in the anteroposterior relationship between the maxillary and mandibular bases (ANB angle), attributed to a slight increase in the SNA angle and a reduction in the SNB angle. However, it is important to note that the improvement in skeletal parameters should not constitute a clinically significant outcome for two reasons: (1) the decrease in the SNB angle was mainly ascribed to the restoration of a proper CO–CR relationship with the backward position of the mandible once the overjet was corrected, and (2) the absence of a control group did not allow to verify whether the increase in the SNA angle was related to the effect of therapy or attributable to a physiological growth process of the premaxilla. Considering the present findings and the lack of evidence on this topic, the use of elastodontic devices or bimaxillary plates should not be recommended with the intention of correcting or influencing skeletal growth patterns in children with class III malocclusion.

In this regard, there is robust evidence that skeletal control of class III malocclusion in growing subjects can be managed by maxillary protraction with a Delaire facemask, a Petit facemask (with or without preliminary maxillary expansion), an SEC III protocol, or functional appliances [15,17,18,19,20]. All these appliances represent the gold standard treatment of skeletal class III malocclusion. Instead, the protocols used in the present study, i.e., elastodontic appliances or bimaxillary plates, can be used in the mild form of class III, including pseudo-class III malocclusion or when the usage of extra-oral or functional appliances is procrastinated. As mentioned above, intra-oral appliances would represent a first line of intervention for treating children with mild class III and one cannot not exclude further intervention with orthopaedic appliances.

In both groups, vestibularization of the upper incisors and retroclination of the lower incisors were observed. However, these data were significantly greater in the PG group compared to the EG group. This difference may be attributed to a greater dentoalveolar effect in the PG group, resulting from increased stress on the upper and lower incisors due to the use of class III elastics. In contrast, in the EG group, dentoalveolar compensations would be less pronounced as the incisors are guided during eruption without the application of orthodontic forces but instead through eruption guides [21]. This finding holds clinical significance, considering that both devices are effective in the short-term interceptive treatment of class III malocclusion, and since it cannot be excluded that these patients may undergo future therapy with orthopedic devices, it may be preferable to use a system in the initial phase that reduces dentoalveolar compensations. Furthermore, considering that some of the patients included in the study had not fully erupted incisors, these findings imply that initiating interceptive treatment during the early stage of incisor eruption is appropriate. Both appliances can serve as guides for eruption, promoting the establishment of normal contact and tipping of the incisors [22].

Regarding the analysis of palate morphology, the surface analysis suggests that the EG group exhibits greater sagittal growth of the palate in the retro-incisive region, as well as increased transverse and vertical growth compared to the PG group. This observation is evident from the color-coded map showing that the area of mismatch between the pre-treatment and post-treatment models is more intense (intense red tone) and extensive in the EG group compared to the PG group (yellow-to-red tones). This result could be interpreted based on the design of the elastodontic device, which isolates the maxilla from the centripetal forces exerted by the peri-oral musculature, favoring a more physiological tongue posture that allows for greater transverse growth. Additionally, the maxillary plate used in the PG group creates interference between the palate vault and the lingual surface, explaining why these subjects exhibited less transverse and vertical growth of the palate.

In light of the present findings, the null hypothesis of the present study was partially accepted. In this regard, both appliances showed similar outcomes considering the skeletal parameters and the restored positive overjet; on the contrary, subjects treated with elastodontic appliances showed a greater and comprehensive change in palatal morphology with a reduced dentoalveolar effect in the frontal area (incisor upper and lower regions). Accordingly, it could be reasonably concluded that both appliances lead to the correction of class III malocclusion, with a predominance of the dentoalveolar component in the PG group and a mixed component (morphological and dentoalveolar) in the EG group.

### Limitations

The absence of an untreated control group is certainly the main limitation of the present study since it would provide evidence for a comparative evaluation of skeletal growth between the treated and control groups. However, considering the clinical relevance of the malocclusion, the definition of an untreated study sample is unlikely due to ethical reasons.

The limited sample size is another limitation, and the present findings should be interpreted with caution until future evidence is available.

## 5. Conclusions

Elastodontic appliances (EAs) and bi-maxillary plates successfully correct class III malocclusions in children. However, elastodontic devices significantly improved the morphology of the palate, both in the transverse and anteroposterior directions. It can reasonably be concluded that elastodontic devices could be successfully used to resolve anterior crossbite by promoting harmonious restoration of maxillary growth.

## Figures and Tables

**Figure 1 children-10-01219-f001:**
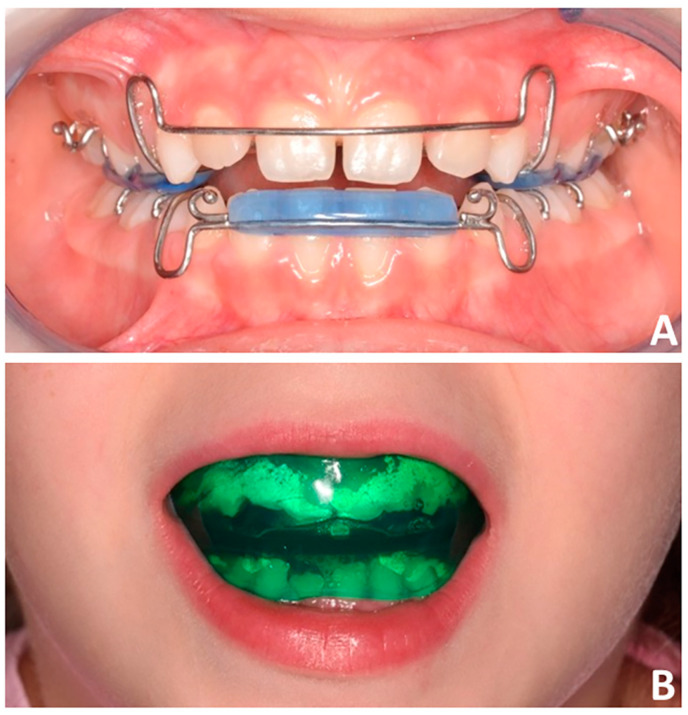
Appliances tested for the purpose of the present study. (**A**) Bi-maxillary plates with class III elastics and (**B**) the class III elastodontic device.

**Figure 2 children-10-01219-f002:**
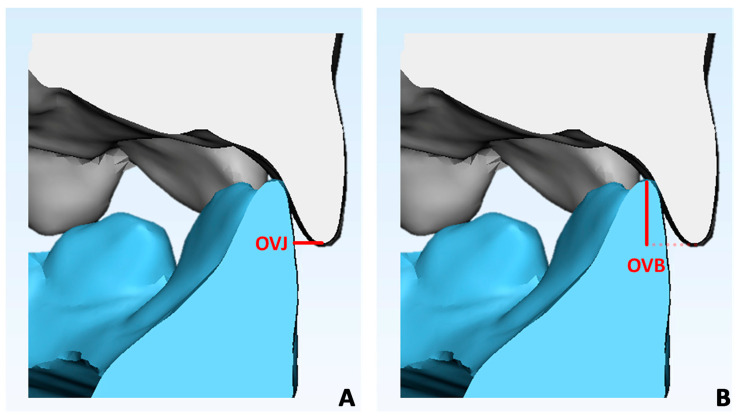
The digital models were imported into 3-Matic software to perform linear occlusal measurements: (**A**) overjet calculation and (**B**) overbite calculation.

**Figure 3 children-10-01219-f003:**
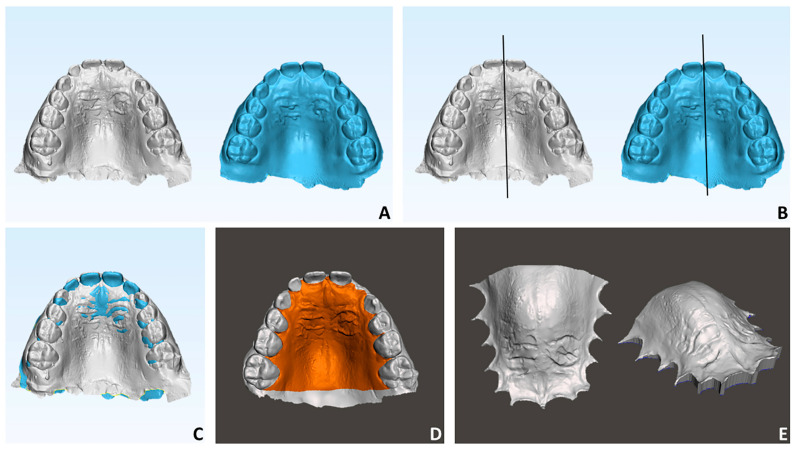
Digital process involved for the surface analysis of the palate. (**A**) T0 (gray) and T1 (light blue) maxillary models (3-Matic software); (**B**) generation of the median palatal plane (MPP) using two points respectively located on the median palatal raphe adjacent to the second ruga and 1 cm distal to the first point (3-Matic software); (**C**) superimposition between T0m and T1 maxillary models using MPP as reference plane (3-Matic software); (**D**) segmentation of the palate excluding dentition (MeshMixer software version 3.5); (**E**) final digital palatal model (MeshMixer software).

**Figure 4 children-10-01219-f004:**
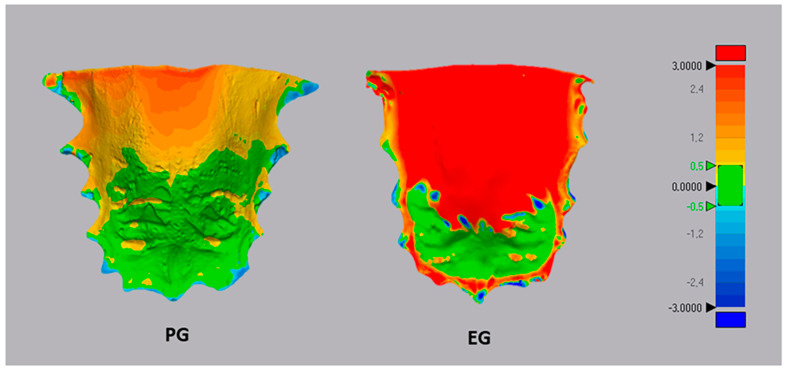
Deviation analysis and calculation of the percentage of matching between T0 and T1 maxillary models for both PG and EG groups. The RGB colored scale bar (millimeters) is shown on the right: the upper (red) and lower (blue) parts of the scale indicate the maximum positive and negative deviations. Green indicates the tolerance range, set to 0.5 mm.

**Figure 5 children-10-01219-f005:**
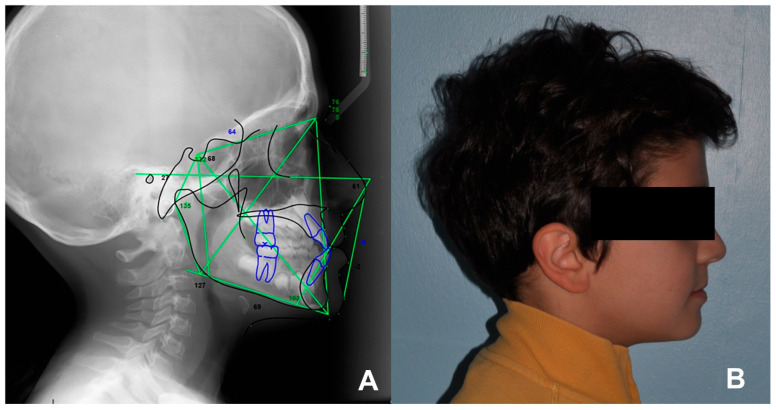
Example of analysis of sagittal discrepancy. (**A**) Cephalometric examination analysis performed on a latero-lateral cephalogram and (**B**) extra-oral right profile photograph.

**Table 1 children-10-01219-t001:** Demographic and clinical characteristics of the sample of the study.

Sample Characteristics	Total (n = 20)	PG (n = 10)	Total (n = 10)	Significance
	Mean/n	Mean/n	Mean/n	
Mean age	7.3 (±1.1)	6.9 (±1)	7.5 (±0.9)	*p* = 0.098
Gender				*p* = 0.132
Male	12	7	5	
Female	8	3	5	
Skeletal Maturity				*p* = 0.074
CVMS 1	18	9	9	
CVMS 2	2	1	1	

*p*-value for comparison of group means by using the *t*-test or differences in the proportion calculated by the chi-square test. CVMS = cervical vertebrae maturation stages.

**Table 2 children-10-01219-t002:** Inferential statistics for the diagnostic parameters assessed via a lateral cephalogram.

	EG Group						PG Group					
	T0		T1				T0		T1			Significance **
	Mean	SD	Mean	SD	Significance *		Mean	SD	Mean	SD	Significance *	
SNA^	78.5	1.21	79.3	1.09	*p* = 0.071	SNA^	77.1	1.16	78.1	1.14	*p* = 0.084	*p* = 0.168
SNB^	79.8	1.07	78	1.11	*p* < 0.05	SNB^	78.6	0.99	76.5	1.03	*p* < 0.05	*p* = 0.211
ANB^	−1.3	0.79	1.3	0.92	*p* < 0.05	ANB^	−1.5	0.85	1.6	0.69	*p* < 0.05	*p* = 0.114
IIA^	139.3	1.4	135.8	1.51	*p* < 0.05	IIA^	138.7	1.35	133.4	1.67	*p* < 0.05	*p* < 0.05
OVJ	−1.4	0.6	1.7	0.5	*p* < 0.05	OVJ	−1.1	0.4	1.8	0.5	*p* < 0.05	*p* = 0.163

T0 = pre-treatment; T1 = post-treatment; SD = standard deviation; IIA^ = interincisal angle; OVJ = overjet. * = *p*-value set at *p* < 0.05 based on paired the Student’s *t* test for inter-timing comparisons; ** = *p*-value based on the independent Student’s *t* test for inter-group comparisons.

**Table 3 children-10-01219-t003:** Comparison of intra-timing matching percentage agreement between pre-treatment (T0) and post-treatment (T1) maxillary models in the PG group and EG group.

	Matching %	SD	Significance
PG Group	59.51	6.03	*p* < 0.05
EG Group	41.07	8.9	

*p*-value set at *p* < 0.05 and based on the independent Student’s *t* test for inter-group comparisons.

## Data Availability

Data will be available upon request to the corresponding author.

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
