# Peer review of "Comparative Analysis of Skeletal Changes, Occlusal Changes, and Palatal Morphology in Children with Mild Class III Malocclusion Treated with Elastodontic Appliances and Bimaxillary Plates"

_children, 2023, doi:10.3390/children10071219_

Round 1

Reviewer 1 Report

Introduction: Add the null hypothesis at the end of the Introduction and then discuss it in the Discussion section. Materials and Methods: What type of study is it? It needs to be mentioned in the manuscript. Materials and Methods: Describe the sample size calculation. Materials and Methods: How was the temporomandibular dysfunction diagnosed? Materials and Methods: Include the manufacturer information of the 3-Matic software. Results: Add the p-values that were non-significant in the tables. Discussion: Enhance the discussion with the results of other studies in the literature. Discussion: Include more references from 2022 and 2023 in the discussion of the results. Limitations: Why do the authors believe that the limited sample size may have influenced the results?"

Author Response

Reply to Reviewer # 1

We warmly thank the reviewer since his considerations/suggestions have significantly increased the quality of our manuscript.

Reviewer’s concern 1. Introduction: Add the null hypothesis at the end of the Introduction and then discuss it in the Discussion section.

Authors’ response. According to the reviewer’s concern, we’ve addressed the null hypothesis and better argue about it in the discussion

Reviewer’s concern 2. Materials and Methods: What type of study is it? It needs to be mentioned in the manuscript.

Authors’ response. According to the reviewer’s concern, we’ve made appropriate changes.

Reviewer’s concern 3. Materials and Methods: Describe the sample size calculation.

Authors’ response. According to the reviewer’s concern, we’ve reported sample size calculation.

Reviewer’s concern 4. How was the temporomandibular dysfunction diagnosed?

Authors’ response. We thank the reviewer for addressing this concern. We should have been clearer. Each patient underwent a through clinical orthodontic and functional evaluation. Subjects were excluded if present TMJ sound or pain. We’ve better specified this in the exclusion criteria. Thank you for the important suggestion.

Reviewer’s concern 6. Materials and Methods: Include the manufacturer information of the 3-Matic software.

Authors’ response. According to the reviewer’s concern, we’ve made appropriate changes.

Reviewer’s concern 7.  Results: Add the p-values that were non-significant in the tables.

Authors’ response. According to the reviewer’s concern, we’ve made appropriate changes

Reviewer’s concern 8. Discussion: Enhance the discussion with the results of other studies in the literature.

Authors’ response. According to the reviewer’s concern, we’ve made appropriate changes. IN particular, we have better argued about the comparison with gold standard treatment for class III such as facemask or functional appliance. There are no studies in literature testing elastodontic appliances for class III malocclusion.

Reviewer’s concern 9. Discussion: Include more references from 2022 and 2023 in the discussion of the results.

Authors’ response. According to the reviewer’s concern, we’ve made appropriate changes. If the reviewer retains that we should provide more references we are willing to to that.

Reviewer’s concern 10. Limitations: Why do the authors believe that the limited sample size may have influenced the results?"

Authors’ response. Our study sample was powered enough to detect differences between the two groups. However, a wider study sample could provide stronger results.

Reviewer 2 Report

I would like to thank the authors for submitting this manuscript for peer review. I found the topic very interesting.

My recommendations to the authors are to correct Table 2 (both groups are EG - none are PG but one of them must be). In addition, I find it difficult to accept the comparisons made between the two appliances without a control group. I would highly recommend that the study be revised to include such a group.

In addition, although it is stated that an IRB approved the study, there is no mention of an approval number - this should be submitted as well.

Minor English grammar improvements are required.

Author Response

Reply to Reviewer # 2

We warmly thank the reviewer since his considerations/suggestions have significantly increased the quality of our manuscript.

Reviewer general considerations. I would like to thank the authors for submitting this manuscript for peer review. I found the topic very interesting.

Authors’ response. We warmly thank the reviewer for the positive response and for having appreciated our efforts in conducting this study.

Reviewer’s concern 1. My recommendations to the authors are to correct Table 2 (both groups are EG - none are PG but one of them must be).

Authors’ response. According to the reviewer’s concern, we’ve made appropriate changes. Sorry, it was a typo error.

Reviewer’s concern 2. In addition, I find it difficult to accept the comparisons made between the two appliances without a control group. I would highly recommend that the study be revised to include such a group.

Authors’ response. We warmly thank the reviewer for addressing this topic. We totally agree with the reviewer, the absence of the control group is the main limitation of the present study. A control group could provide information about changes occurring due to growth and compare them from those obtained due to treatment. However, considering the importance of treating class III malocclusion in growing subject, it is quite difficult to have a control group of untreated subjects due to ethical issues. As consequence, we cannot provide a control group. However, we retain that we have provided useful information in the comparison between two treatment protocols.

Reviewer’s concern 3. In addition, although it is stated that an IRB approved the study, there is no mention of an approval number - this should be submitted as well.

Authors’ response. According to the reviewer’s concern, we’ve made appropriate changes. Sorry, we’ve forgotten to provide approval number. We’ve now reported such information.

Round 2

Reviewer 2 Report

To the authors:

I have reviewed your revisions and have read your responses.

I accept your argument that a control group would be difficult to justify ethically.

Please correct some minor grammatical errors in the manuscript.